# Bell-Type Correlation at Quantum Phase Transitions in Spin-1 Chain

**DOI:** 10.3390/e22111282

**Published:** 2020-11-12

**Authors:** Dongkeun Lee, Wonmin Son

**Affiliations:** 1Department of Physics, Sogang University, 35, Baekbeom-ro, Mapo-gu, Seoul 04107, Korea; dglee90@sogang.ac.kr; 2Research Institute for Basic Science, Sogang University, 35, Baekbeom-ro, Mapo-gu, Seoul 04107, Korea

**Keywords:** bell-nonlocality, quantum phase transition, many-body systems

## Abstract

For the identification of non-trivial quantum phase, we exploit a Bell-type correlation that is applied to the one-dimensional spin-1 XXZ chain. It is found that our generalization of bipartite Bell correlation can take a decomposed form of transverse spin correlation together with high-order terms. The formulation of the density-matrix renormalisation group is utilized to obtain the ground state of a given Hamiltonian with non-trivial phase. Subsequently Bell-type correlation is evaluated through the analysis of the matrix product state. Diverse classes of quantum phase transitions in the spin-1 model are identified precisely through the evaluation of the first and the second moments of the generalized Bell correlations. The role of high-order terms in the criticality has been identified and their physical implications for the quantum phase have been revealed.

## 1. Introduction

Quantum correlation, that has no classical counterparts, plays a pivotal role in studying the many-body systems. Investigation of such a complex massive system through correlation is a major research topic in statistical and condensed matter physics as they provide useful views for the quantification of physical properties in large scale systems [1]. Especially, the trait of quantum phase and its phase transition driven by the quantum fluctuation in correlation have attracted much interest throughout the years as they are closely related to macroscopic quantum phenomena, such as superconductivities and topological state of a matter. They are analyzed through the various measures that have been rooted in quantum information theory.

Quantum entanglement is one of the most crucial concepts that has been extensively studied in the last two decades to comprehend the quantum phase transitions and the exotic quantum properties in condensed matter systems [2]. Concurrence, one of the most used measures for two-qubit entanglement, is exploited to estimate the quantum criticality and, further, its scaling behavior is also analyzed near the critical point of the one-dimensional XY model [3,4]. Entanglement entropy of a block in various many-body systems has also been discussed during the analysis using conformal field theory [5]. The entanglement entropy demonstrates logarithmic divergence at critical points, whereas it saturates noncritical systems associated with the area law [5,6]. The localizable entanglement, another measure of entanglement, is used to detect the phase transition in spin-1/2 XXZ model and to show the divergence of the entanglement length in valence-bond solid states [7,8]. Subsequently, entanglement instabilities demonstrates the origin of the quantum criticality precisely and their analytic dynamics as the interaction parameters are modified [9].

It is well known that all pure entangled states for two-qubit systems violate CHSH-Bell inequality [10]. On the other hand, the nonlocal character of mixed entangled states are not easily quantifiable since it does not necessarily violate Bell inequalities [11]. It means that quantum entanglement is not necessarily equivalent to the concept of Bell nonlocality [11,12]. Thus, it can be said that the concept of entanglement is closely related to nonlocality while they addresses the different physical quantities in general.

Unlike many-body entanglement, that has been subjected to extensive investigation, the Bell nonlocality in many-body systems has been attracted less attention relatively so far. It is because generalization of nonlocality is needed to consider all the possible local measurements and locate their nontrivial combinations of the joint probabilities. It is only recent that the nonlocality in many-body systems gets to make an active progress. At first, Bell correlations for many-body systems are applied to two-site reduced density states which are generated from the ground state in one-dimensional (1D) spin-1/2 XY model [13] and spin-1/2 XXZ model [14]. As a result, it has been known that the Bell correlation demonstrates the non-analyticity at the critical point while, due to the monogamy characteristics of the correlation, any bipartite Bell inequality is not violated by the translational invariant many-body systems [15].

It is in two-dimensional local Hilbert space that the most Bell correlations are elucidated in aforementioned studies. In other words, Bell correlations in *d*-dimensional systems of many particles have been rarely investigated, whereas the other measures of quantum information theory, such as fidelity and quantum coherence, are used to characterize the quantum phase transitions (QPTs) many times in spin-1 chains or more [16,17,18]. In this work, we use a generalized high-order Bell correlation obtained from high-dimensional spin measurements, which is proposed by D. Collins et al. [19] and W. Son et al. [20,21]. Then, we analyze its Bell correlation in the spin-1 chain to identify diverse quantum phase transitions. Moreover, we discuss whether the violation of the Collins–Gisin–Linden–Massar–Popescu (CGLMP) inequality can be detected even in three-dimensional systems.

In particular, we choose 1D spin-1 XXZ model with the onsite anisotropy, which is one of the most studied spin-1 chain models. Higher-spin XXZ systems have richer quantum phases in various different parameter regimes and certain parts of the system are need to be depicted with more complicated physics, such as symmetry-protected topological phases [22]. Since the ground state of spin-1 chains is not exactly obtainable, we utilize a numerical technique—the infinite-size density-matrix renormalization group (iDMRG) method—for the ground state of spin-1 system, which is described in matrix product state (MPS) representations.

The outline of this paper is as following: In Section 2, we start with the system hamiltonian of the 1D spin-1 XXZ model with the onsite anisotropy. There, we briefly explain its phases at the different parameter regime and their criticalities at the QPTs. In Section 3, we show the Bell correlation in the CGLMP inequality from the generalized SLK–Mermin–Bell formalism described in an operator form. Then, using the state obtained, we present our results for the description of Bell correlations in overall parameter space and provide a close analysis in the vicinity of the criticality in Section 4, and we conclude this paper in Section 5.

## 2. The 1d Spin-1 XXZ Model with the On-Site Anisotropy

We now start with the description of the system to be analyzed. Our choice of model is made on the system with the versatile quantum phases occurred by the second-order magnetization term. The Hamiltonian for the one-dimensional spin-1 XXZ model with the on-site anisotropy is in the form of
(1)$$\begin{array}{c} \hat{H}=\sum_{j}J\left({\hat{S}}_{j}^{x}{\hat{S}}_{j+1}^{x}+{\hat{S}}_{j}^{y}{\hat{S}}_{j+1}^{y}+J_{z}{\hat{S}}_{j}^{z}{\hat{S}}_{j+1}^{z}\right)+D\sum_{j}{\left({\hat{S}}_{j}^{z}\right)}^{2}, \end{array}$$
where ${\hat{S}}_{j}^{a}$ for a=x,y,z denotes the spin-1 operators at site *j*. The summation in Equation (1) illustrates the one-dimensional spin-1 XXZ chain with the exchange anisotropy $J_{z}$ and we set J=1 for the unit energy scale. The second sum in Equation (1) describes an on-site anisotropy where the constant *D* means the uniaxial single-site anisotropy.

It has been known that this model can be found in various phases, including the antiferromagnetic (AFM), the ferromagnetic (FM), XY1, XY2, Haldane, and the large-D phases. The FM and AFM phases have the magnetic order. In the former, all spins point in the same direction and, in the latter, spins at the nearest neighbor sites, are aligned in the opposite directions. The XY phase is that all spins, which interacts with the nearest neighbors, which lie in the xy plane and are not allowed to point in the spin-*z* direction. The XY models are gapless, so that the spin–spin correlation $\langle {\hat{S}}_{i}^{+}{\hat{S}}_{i+r}^{-}\rangle$ and $\langle {\left({\hat{S}}_{i}^{+}\right)}^{2}{\left({\hat{S}}_{i+r}^{-}\right)}^{2}\rangle$ display a power-law decay for XY1 and XY2 phases of Equation (1), respectively [23,24,25]. For Haldane and large-D phases, the system has a finite gap and exponentially decaying spin–spin correlations [26,27]. The string order parameter, the expectation value of the nonlocal operator (here, the term, nonlocal, is not the same as the Bell nonlocality, meaning the violation of Bell inequality—see more Ref [28]), has nonzero values in the Haldane phase, when the system preserves the $Z_{2}\times Z_{2}$ symmetry [29,30,31]. Later, the Haldane and large-D phases can be distinguished by the symmetry-protected topological order [32,33,34,35].

Additionally, as the parameters $J_{z}$ and *D* vary, the ground state of the system undergoes diverse types of QPTs, such as Ising-type, Berezinskii, Kosterlitz, and Thouless (BKT) phase transitions [23]. The ground-state phase diagram of the spin-1 chain Equation (1) is investigated in an earlier study [36]. Here, we briefly demonstrate the characteristics of the different phases and QPTs in the given parameter space.

## 3. Bell-Type Correlation from the Generalized Nonlocality Criteria

Bell-type inequalities examine the correlation between two parties of a composite quantum state and determine whether a given quantum state is nonlocal through the violation of local realistic bounds. The Clauser–Horne–Shimony–Holt (CHSH) version of Bell inequality is given for the simplest symmetric states where two dichotomic measurements are performed for each of the parties [37]. If the CHSH correlation admits the local hidden variable model, its absolute value does not exceed the local realistic bound 2. Taking pure entangled states into account, one can find violation of the CHSH inequality and its maximal violation $2\sqrt{2}$ derived by Tsirelson [38]. For the case of two-qubit mixed states, entangled ones do not always violate this inequality [11].

When it comes to the systems of arbitrary dimension *d*, the inequality derived by Collins et al. provides a generalized version of CHSH inequality [19]. Further generalization of the CHSH inequality for the multiparty *d*-dimensional systems has been made by Son et al., which is also consistent with the CGLMP inequality [20,21]. In these scenarios, all the parties are permitted to perform two different measurements at each site with *d*-outcome measurements.

From here, the correlation on the left hand side of the inequality for d×d system is said to be *CGLMP correlation* as it reduced into the one found by CGMLP [19]. Our description is made not by a set of probabilities but by the form of correlation functions. It can be proved that the representations are equivalent as long as an appropriate collection of the correlations is made through the right choices of weighting coefficients [21].

In the scenario of the CGLMP inequality, there are two observers (j=1,2) and each of them has two measurement choices $V_{j}\in \left\{A_{j},B_{j}\right\}$. The measurement operators for the *j*-th party are given as
(2)$${\hat{V}}_{j}\equiv \sum_{\alpha =0}^{d-1}\omega^{\alpha }{|\alpha \rangle }_{V_{j}}\langle \alpha |,$$
where *d* is the index of the number of possible outcomes, $\omega^{\alpha }$ is the eigenvalue, and ${|\alpha \rangle }_{V_{j}}$ is the corresponding eigenvector. One observer chooses the Fourier transformed basis and the other observer takes the inverse Fourier basis. The Fourier transformed basis is given as
(3)$$\begin{array}{c} {|\alpha \rangle }_{V_{j}}=\frac{1}{\sqrt{d}}\sum_{\beta =0}^{d-1}\omega^{-(\alpha +\phi_{V_{j}})\beta }|\beta \rangle , \end{array}$$
where ω=exp(2πi/d) and $\phi_{V_{j}}$ denotes a phase shift for a measurement ${\hat{V}}_{j}$. Here, the basis |β〉 is nothing but the computational basis, such as the spin-*z* basis for d=2.

As for maximal tests, four different measurement choices can be made through four different phase shifts that are $\phi_{A_{1}}=0$, $\phi_{B_{1}}=1/2$, $\phi_{A_{2}}=-1/4$ and $\phi_{B_{2}}=1/4$. Based upon this expression, CGLMP correlation can be described by the expectation value of the following composite operator,
(4)$$\begin{array}{cc} \langle \hat{B}\rangle =\langle \sum_{n=1}^{d-1}f_{n}\left({\hat{A}}_{1}^{n}+\omega^{n/2}{\hat{B}}_{1}^{n}\right)⊗\left({\hat{A}}_{2}^{n}\right. & +{\left.\omega^{n/2}{\hat{B}}_{2}^{n}\right)}^{†}\rangle +c.c. \end{array}$$
when the weight function becomes $f_{n}\equiv \frac{1/2}{d-1}\omega^{\frac{n}{4}}\sec\left(n\pi /2d\right)$. As addressed in [21], the weight $f_{n}$ can be specified for the convex sum of the probability distributions suggested by Collins et al. [19]. Bell correlation Equation (4) for the case of d=2 is obviously equivalent to one in the CHSH inequality. The local realistic bounds of the CGLMP correlation for d=3 are given by $-4\leq \langle \hat{B}\rangle \leq 2$, which can be derived from $f_{n}$ [21].

It is convenient to investigate the structure of the operator $\hat{B}$ in Equation (4) by introducing the *n*-level lowering operator ${\hat{J}}_{j}^{n}\equiv \sum_{\beta =n}^{d-1}{|\beta \rangle }_{j}\langle \beta -n|$ for *j*-th party. From Equations (2) and (3), one can derive the identities such that
(5)$$\frac{1}{2}\left({\hat{A}}_{1}^{n}+\omega^{n/2}{\hat{B}}_{1}^{n}\right)={\hat{J}}_{1}^{n},\;\;\;\;\frac{1}{2}\left({\hat{A}}_{2}^{n}+\omega^{n/2}{\hat{B}}_{2}^{n}\right)=\omega^{n/4}{\hat{J}}_{2}^{n}$$
where the different choice of measurements on the second system, as shown in Equation (4), results in the extra shift of the local phase. Thus, together with the replacement, the Bell operator $\hat{B}$ can be rewritten in a simplified manner as
(6)$$\begin{array}{c} \hat{B}=\frac{2}{d-1}\sum_{n=1}^{d-1}\sec\left[\frac{n\pi }{2d}\right]\left({\hat{J}}_{1}^{n}⊗{\hat{J}}_{2}^{n†}+{\hat{J}}_{1}^{n†}⊗{\hat{J}}_{2}^{n}\right). \end{array}$$

The operator can be applied to the system with the local basis dimension *d*. In this work, we analyze the spin-1 chain where the dimension of local basis is set to be three—i.e., d=3. When the set of states {|β〉|β=0,1,2} is regarded as the computational basis, the matrix representation of $\hat{J}$ and ${\hat{J}}^{2}$ is given as
(7)$$\begin{array}{c} \hat{J}=\left(\begin{array}{ccc} 0 & 0 & 0\\ 1 & 0 & 0\\ 0 & 1 & 0 \end{array}\right)\;\;\;\mathrm{and}\;\;\;{\hat{J}}^{2}=\left(\begin{array}{ccc} 0 & 0 & 0\\ 0 & 0 & 0\\ 1 & 0 & 0 \end{array}\right). \end{array}$$

The operators shift the higher spin state into the lower spin state in one step and two steps, respectively. The choice of measurements $A_{i}$ and $B_{i}$ in Equation (4) using the corresponding $\phi_{A_{i}}$ and $\phi_{B_{i}}$ leads to the lowering operators of higher order and they identifies all the matrix elements of the given state. It can be found that the convex sum of the correlation function provides the symmetric selection of relevant probabilities for the correlations in the parties as addressed in [21].

## 4. Quantum Criticality through the Correlation for the Nonlocality Tests

First, let us present the CGLMP correlation $\langle {\hat{B}}_{i,i+r}\rangle$ that is obtained by the expectation value of the operator $\hat{B}$ for the ground state of Equation (1) in the thermodynamic limit. A quantum state |Ψ〉 on the 1D translationally invariant systems of length *N* can be written in the MPS form [39,40,41,42]
(8)$$\begin{array}{c} |\Psi \rangle =\sum_{s_{1},\ldots ,s_{N}}\Gamma_{s_{1}}\Lambda \Gamma_{s_{2}}\ldots \Lambda \Gamma_{s_{N}}|s_{1}\ldots s_{N}\rangle , \end{array}$$
where $\Gamma_{s_{j}}$ is the χ×χ matrix with the bond dimension χ and $|s_{j}\rangle$ is the *j*th-site local basis of dimension *d* with the physical indices $s_{j}=0,\ldots ,d-1$. Here, the class of matrices $\Gamma_{s_{j}}$ and Λ for the ground state can be obtained from the iDMRG method (we simulate the iDMRG method and calculate some expectation values by using TeNPy Library (version 0.4.1) [43]. We choose the bond dimension χ=200 and specially confine a parameter for convergence such that the relative difference of entanglement entropy in each sweep is not over $10^{-7}$.).

Here we obtain the characteristics of the CGLMP correlation for the particular set of fixed parameters D=0, $J_{z}=1$ and $J_{z}=-1$ in Equation (1). In the parameter region, various phases and QPTs have been identified through the criticality of Bell correlations although the quantum violation of the CGLMP inequality cannot be observed. They are presented in Figure 1a, Figure 2a and Figure 3a. Such nonviolation of the CGMLP inequality is consistent with the cases of the CHSH inequality in the earlier studies of 1D spin-1/2 XY chain [13] and XXZ chain [14].

Nevertheless, the CGLMP correlation $\langle {\hat{B}}_{r}\rangle$ can be applied to elucidate the physical properties for the various quantum phases in the system Equation (1). Criticality can be clearly found through the non-analyticity of the CGLMP correlation. Moreover, it is notable that the contribution of higher-order correlation for the criticality is possible to be discriminated in our formalism. The noble feature of the rationales is that the versatile quantification of quantum phases is possible to be identified through them. It is since Equation (6) that d=3 can be decomposed into a linear combination of spin–spin correlation functions in the first and the second order moments where the *n*-level lowering operator ${\hat{J}}_{j}^{n}$ is regarded as the spin-1 lowering operator ${\left({\hat{S}}^{-}\right)}_{j}^{n}$—i.e., ${\hat{J}}^{†}\equiv {\hat{S}}^{+}/\sqrt{2}$ and $\hat{J}\equiv {\hat{S}}^{-}/\sqrt{2}$. It is, thus, given by
(9)$$\begin{array}{c} \langle {\hat{B}}_{r}\rangle =2\left(\frac{1}{\sqrt{3}}\;C_{r}^{\left(1\right)}+\frac{1}{2}C_{r}^{\left(2\right)}\right), \end{array}$$
where the transverse spin–spin correlations $C_{r}^{\left(1\right)}$ and $C_{r}^{\left(2\right)}$ are given by
(10)$$\begin{array}{c} C_{r}^{\left(1\right)}\equiv \langle {\hat{S}}_{i}^{+}{\hat{S}}_{i+r}^{-}\rangle \;\;\;and\;\;\;C_{r}^{\left(2\right)}\equiv \langle {\left({\hat{S}}_{i}^{+}\right)}^{2}{\left({\hat{S}}_{i+r}^{-}\right)}^{2}\rangle . \end{array}$$

In Equation (9), we drop the index *i* because these correlation functions depend not on the site *i* but on the relative distance *r* between two spins due to the translational symmetry in the thermodynamic limit. It is known that the qualitative forms of the correlation functions $C_{r}^{\left(1\right)}$ and $C_{r}^{\left(2\right)}$, including various exponents, can be expected by using the bosonization technique, an analytical approach based on the effective field theory with defects [23].

Thus, we discuss below how the CGLMP correlation Equation (9) can be used as an indicator of the QPTs by identifying the extremum and inflection points [44]. It is also discussed how the Bell-type correlations vary as the distance *r* changes, especially at the point of QPT, in the following subsections.

### 4.1. The CGLMP Correlation at D = 0

Let us analyze our evaluation of the CGLMP correlation at D=0 as varying the anisotropic exchange interaction $J_{z}$. The CGLMP correlation is zero in the FM phase ($J_{z}<-1.0$), which is trivially illustrated. A discontinuous change of $\langle {\hat{B}}_{r}\rangle$ can be detected at $J_{z}=-1$ where the first-order QPT occurs at the point. The spins in the AFM phase ($J_{z}≳1.18$) are inclined to locate the spins alternatively in either |+〉 or |−〉 site by site. A very large $J_{z}$ gives rise to nearly vanishing value of the transverse spin–spin correlations and the CGLMP correlation becomes consequently close to zero. The inflection point of the CGLMP correlation, $\partial^{2}\langle \hat{B}\rangle /\partial J_{z}^{2}$=0, can be found at $J_{z}=1.18$, which turns the AFM phase into different phases. This type of transition is similar to the criticality in the model of two-dimensional Ising interaction. Moreover, the critical point $J_{z}=1.18$ agrees with known results from the previous analysis using the DMRG techniques [18,45,46].

The crucial region for the non-trivial phase lies at the interval between the FM and the AFM phases. For spin-1/2 system, only the XY phase exists in this interval, while there are the XY phase and the Haldane phase for the spin-1 systems. Therefore, more phase transition points exist between the XY phase and the Haldane phase whereas the transition type is going to be the BKT type QPT [23]. However, both $C_{r}^{\left(1\right)}$ and $C_{r}^{\left(2\right)}$ do not show the critical behavior solely in the region because minimum or the inflection points do not appear, as can be seen in Figure 1b,c. Therefore, the CGLMP correlation does not identify the BKT-type QPTs near the point $J_{z}=0$ at which the QPT in the spin-1 XXZ chain is expected to occur [36]. Instead of identifying the transition point from the CGLMP correlation directly, the BKT-type QPT can be indicated by an indirect manner, such as vanishing string order parameter, or a local maximum of the entanglement entropy [45,46]. In this case, it is demonstrated that the correlation function $C_{r}^{\left(1\right)}$ evaluated from iDMRG method fits a function $ar^{\eta }+b$ near $J_{z}=0$. Through the extrapolation, the critical exponents $\eta_{c}=1/4$ of the BKT-type QPT can be found at $J_{z}=0.02$.

In comparison to the spin-1 case, it is intriguing that BKT-type QPT can be identified by the CHSH correlation in the spin-1/2 XXZ chain [14]. At $J_{z}=1$, the spin-1/2 XXZ chain undergoes BKT-type QPT and has the SU(2) symmetry. It is possible that the SU(2) symmetry, rather than criticality, is involved in the minimum of the CHSH correlation for $J_{z}=1$. To be more specific, while $\langle {\hat{S}}_{i}^{x}{\hat{S}}_{i+r}^{x}\rangle <\langle {\hat{S}}_{i}^{z}{\hat{S}}_{i+r}^{z}\rangle$ in the AFM phase ($J_{z}>1$), $\langle {\hat{S}}_{i}^{x}{\hat{S}}_{i+r}^{x}\rangle >\langle {\hat{S}}_{i}^{z}{\hat{S}}_{i+r}^{z}\rangle$ in the XY phase ($J_{z}<1$). Due to the SU(2) symmetry, $\langle {\hat{S}}_{i}^{x}{\hat{S}}_{i+r}^{x}\rangle =\langle {\hat{S}}_{i}^{z}{\hat{S}}_{i+r}^{z}\rangle$ for $J_{z}=1$, which affects the maximization of the CHSH correlation with respect to the measurement directions in each phase.

Let us discuss the CGLMP correlation $\langle {\hat{B}}_{r}\rangle$ as a function of the distance *r* between two sites. We expect that the decaying behavior of the CGLMP correlations are affected by that of the spin–spin correlations $C_{r}^{\left(1\right)}$ and $C_{r}^{\left(2\right)}$. The spin–spin correlation $C_{r}^{\left(1\right)}$, rather than $C_{r}^{\left(2\right)}$ in Figure 1b, dominates $\langle {\hat{B}}_{r}\rangle$ without the onsite anisotropy. Then, the CGLMP correlation as a function of *r* exhibits the even–odd oscillation. A notable feature of the CGLMP correlation is shown in the XY1 phase: $\langle {\hat{B}}_{r}\rangle$ still has a finite value due to a power-law decay of $C_{r}^{\left(1\right)}$, whereas in other phases it drastically approaches to zero at a large distance *r* (see Figure 1). For the Haldane phase and the AFM phase, the CGLMP correlations $\langle {\hat{B}}_{r}\rangle$ are also characterized by an exponential decay, which results from the behavior of the spin–spin correlations $C_{r}^{\left(1\right)}$ and $C_{r}^{\left(2\right)}$. In Figure 1a, CGLMP correlation in the Haldane phase decays less slowly than the AFM phase. These features correspond to the fact that the spin–spin correlation function decays exponentially in the AFM phase and one also has an exponential decaying with an additional factor $\sqrt{r}$ in the Haldane phase—i.e., $C_{r}^{\left(1\right)}∼{\left(-1\right)}^{r}r^{-1/2}\exp\left(-r/\xi \right)$ at the Heisenberg point ($J_{z}=1$ and D=0) [27,47].

### 4.2. The CGLMP Correlation at $J_{z}$ = 1

The Hamiltonian in Equation (1) goes through three different phases by tuning the single-ion anisotropy *D* at $J_{z}=1$—the AFM, the Haldane, and the large-D phase. In addition, there are two types of QPTs that correspond to an extremum point (either minimum for an odd *r* or maximum for an even *r*) and an inflection point of the CGLMP correlation $\langle {\hat{B}}_{r}\rangle$ for each *r* in Figure 2a. These critical points are influenced by ones for both correlations $C_{r}^{\left(1\right)}$ and $C_{r}^{\left(2\right)}$. In the previous section, the characteristics of the Haldane and the AFM phase have been discussed and for $J_{z}=1$ we can also detect the Ising-type QPT at $D_{c_{1}}≃-0.31$ between them, a peak of $\partial \langle {\hat{B}}_{r}\rangle /\partial D$ for small *r* compared with the fidelity [16]. So, we are going to concentrate on the characteristics of the Haldane phase, and its relationship with the large-D phase. It is noteworthy in Figure 2a that the phase transition lies between the Haldane phase and the large-D phase, often called the Gaussian phase transition.

The string order parameter, consisting of spin operators, has a nonzero value only in the Haldane phase and thereby is widely used to discriminate the Haldane and large-D phase in this model. Later, the Haldane phase and the large-D phase are featured by a nontrivial topological phase and a trivial one under the symmetry protection, respectively [32,35]. It is suggested that the order parameter which reflects the symmetry-protected topological order in the presence of $Z_{2}\times Z_{2}$ symmetry becomes −1 for the Haldane and 1 for the large-D phase [35]. Furthermore, other various studies have revealed the boundary of the Haldane and large-D phases. The structure for the energy levels has been mainly analyzed to capture the accurate transition point $D_{c_{2}}=0.975$ [46]. Additionally, the behaviors of the ground-energy state have been treated by investigating the fidelity ($D_{c_{2}}=0.97$) [16] and the entanglement entropy ($D_{c_{2}}=0.96845\left(8\right)$) [48], which all use the DMRG method.

In our simulation (see Figure 2a), $D_{c_{2}}$ the minimum points of $\langle {\hat{B}}_{r}\rangle$ which come from those of spin–spin correlations $C_{r}^{\left(1\right)}$ and $C_{r}^{\left(2\right)}$ are captured around the phase transition between the Haldane and the large-D phase. For r=1, we can find the critical point $D_{c_{2}}=0.914$ by evaluating the minimum of $\langle {\hat{B}}_{r=1}\rangle$. This critical point moves to $D_{c_{2}}=0.968$ at r=81 using the linear interpolation near the criticality in Figure 2c. This critical point converges to $D_{c_{2}}=0.969$ until the CGLMP correlation decays to zero as the distance *r* goes to infinity in Figure 2c. Thus, $\langle {\hat{B}}_{r}\rangle$ deserves an indicator of the QPT for sufficiently distant parties. It is also remarkable that the peak shape in Figure 2a remains in the vicinity of the criticality, even at a large distance *r*. Specifically, the CGLMP correlation $\langle {\hat{B}}_{r}\rangle$ as a function of *r* decays more slowly in the vicinity of the critical point than in the noncritical region. It is because the correlation functions $C_{r}^{\left(1\right)}$ and $C_{r}^{\left(2\right)}$ decay as a power law near the criticality, whereas they decay exponentially in the Haldane and the large-D phase [23,30].

### 4.3. The CGLMP Correlation at $J_{z}=-$0.1

The spin-1 Hamiltonian Equation (1) involves two different types of XY phases, the XY1 phase and XY2 phases [23]. The former which we have been discuss above is located at small *D* and the latter lies along the large negative *D*.

In order to investigate the CGLMP correlation in two types of XY phases and their transition, let us take the anisotropy $J_{z}$ to be −0.1 and change the on-site anisotropy *D*. In Figure 3a,b, the fact that the correlation $C_{r}^{\left(2\right)}$ is more dominant that $C_{r}^{\left(1\right)}$ brings about the positive value of the CGLMP correlation in the XY2 phase ($D<D_{c}$). It is also shown that for large negative *D*, decay of the correlation function $C_{r}^{\left(2\right)}$ obeys power laws while the correlation $C_{r}^{\left(1\right)}$ decays exponentially, which coincides with the XY2 phase [23]. In Figure 3c, we can detect the Ising-type criticality between the XY1 and XY2 phases $D_{c}≃-2.10$ by solving $\partial^{2}\langle {\hat{B}}_{r}\rangle /\partial D^{2}=0$. The farther the distance *r* is, the deeper the peak of $\partial \langle {\hat{B}}_{r}\rangle /\partial D$ is in the vicinity of this criticality. This detection quite exactly corresponds to the level crossing point of two energy gaps, one of which consists of the excitation (or the quantum number) $M^{z}=\pm 2$ in the XY2 phase and the other has $M^{z}=\pm 1$ in the XY1 phase [36].

## 5. Conclusions

We have investigated CGLMP correlations in the 1D XXZ model with onsite anisotropy especially near the quantum phase transitions. In the case of spin-1, the CGLMP correlation obtained from local measurements in the Fourier bases can be interestingly interpreted as a linear combination of the first-order and second-order transverse spin–spin correlations. As a result, we found various QPTs that are accurately indicated by analyzing the derivatives of the CGLMP correlation. The first-order QPT (FM-to-XY transitions) shows discontinuity of the CGLMP correlation. The Ising-type (Haldane-to-AFM and XY1-to-XY2) QPTs occur where the second derivative of the CGLMP correlation vanishes. The Haldane-to-large-D transition, also referred to as the Gaussian phase transition, emerges when $\partial \langle {\hat{B}}_{r}\rangle /\partial D=0$. The criticality of the CGLMP correlation is caused by the fact that both the derivatives of the spin–spin correlations $C_{r}^{\left(1\right)}$ and $C_{r}^{\left(2\right)}$ are simultaneously zero in the criticality. Moreover, decaying behaviors of the CGLMP correlation with respect to the distance *r* in each phase can be predicted from the field-theoretical approach [23]. However, the BKT-type QPT at $J_{z}=0$ cannot be indicated by finding critical points of the derivatives of the CGLMP correlation. This is in contrast to the CHSH correlation in spin-1/2 XXZ chain [14].

Another crucial result is that there is unfortunately nonviolation of the CGLMP inequality in all phases of spin-1 chain; that is, the CGLMP correlations for the ground states of spin-1 chain do not surpass the local realistic bounds. Even though we do not optimize the measurements for the ground states in each phase, this nonviolation coincides with the one in the 1D spin-1/2 XY chain [13] and XXZ chain [14] invariant under translational symmetry. The reduced density matrices on two sites, which display mixed entangled states, are not sufficient to violate the CGLMP inequality in the spin-1 chain, although many-body ground states are pure and highly entangled in this model. As the CHSH correlation has monogamy trade-off relation [15], monogamy of the CGLMP correlation can be a candidate to explain the non-violation of CGLMP inequality in the spin-1 chain. Therefore, in order to detect nonlocality in this model, one should consider multipartite Bell correlations for arbitrary dimensions. This gets a clue from the result that multipartite Bell correlations for spin-1/2 invariant under the permutational symmetry [49] do not satisfy the local hidden variable model in the Ising model with long-range interactions [50,51]. We leave that for future investigation, which is quite consistent with the current approach demonstrated in this work.

## Figures and Tables

**Figure 1 entropy-22-01282-f001:**
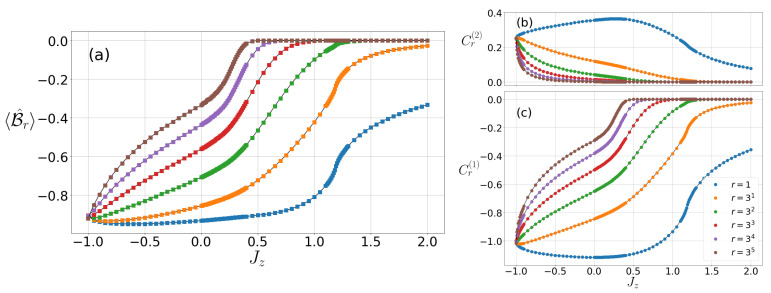
(Color online) (**a**) the CGLMP correlation $\langle {\hat{B}}_{r}\rangle$ and (**b**) the spin–spin correlations $C_{r}^{\left(2\right)}$ and (**c**) $C_{r}^{\left(1\right)}$ as varying the parameter $J_{z}$ at D=0 for odd distances.

**Figure 2 entropy-22-01282-f002:**
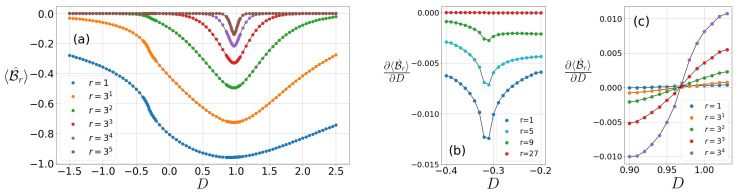
(**a**) CGLMP correlation $\langle {\hat{B}}_{r}\rangle$ and the first derivative of $\langle {\hat{B}}_{r}\rangle$ with respect to *D* near (**b**) D=−0.31 and (**c**) D=0.97 as varying the parameter *D* at $J_{z}=1$ for odd distances.

**Figure 3 entropy-22-01282-f003:**
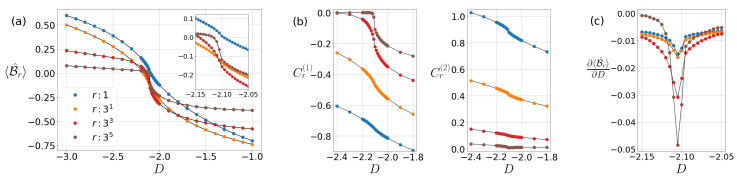
(**a**) CGLMP correlation $\langle {\hat{B}}_{r}\rangle$, (**b**) spin–spin correlations $C_{r}^{\left(1\right)}$ and $C_{r}^{\left(2\right)}$ as varying the parameter *D*, and (**c**) first derivative of $\langle {\hat{B}}_{r}\rangle$ with respect to *D* at $J_{z}=-0.1$ for odd distances *r*.

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
