# Peer review of "Bell-Type Correlation at Quantum Phase Transitions in Spin-1 Chain"

_entropy, 2020, doi:10.3390/e22111282_

Round 1

Reviewer 1 Report

The authors employ density-matrix renormalisation group (DMRG) to study the ground state in the one-dimensional spin-1 XXZ chain which It is well know be characterized by a gap in the spectrum, the Haldane Gap. The non-trivial phase characterized by Bell-SLK-type generalized correlation are evaluated through the analysis of the matrix product state. Athough, the different phases of the spin-1 XXZ model already are many studied in the literature using different analytical techniques of Quantum field theory such as nonlinear Sigma model and numerical techniques such model, Quantum Monte Carlo and so on, (that are not properly cited by the authors), the influence of the critical points on quantum information properties as correlation and quantum entanglement,... are an important subject in nowadays and I think that the results of the present manuscript can be published after a minor revision in the literature.

I think the manuscript can also be accepted in its current form.

Author Response

We appreciate the review by the first referee who had located the importance of our work appropriately. According to the suggestion provided, we had corrected our manuscript as to improve the presentation of our results. The list of changes were itemised in the separate sheet and we wish that our final version can be read more with ease. 

Reviewer 2 Report

The paper "Bell-type correlation at quantum phase transitions in spin-1 chain" addresses the question of the
magnetic phase diagram of 1D spin-1 XXZ chain. The ground state of the system has been studied numerically by DMRG method which is
is essentially the same as that used in the Ref. [28], in which the phase diagram of the model had been found for the first time.
Although the problem (albeit it has been actually solved in Ref. [28], whereas the Bell-type correlations, though in the model
without anisotropy, had been considered in Ref.[23], see also earlier work [12]) is interesting by itself,
I consider the present manuscript to be of minor novelty. This is because all its components have been worked out earlier: the XXZ chain phase diagram in Ref. [28] of the manuscript, while quantum correlations (nonlocality) in the form of Bell inequalities in the above papers. What can be considered novel, is minor corrections to the ground state phase diagram of Ref. [28] due to CGLMP correlation effects. This already makes the manuscript to be unsuitable for publication in the Entropy.

Minor remark is that the English is inadequate in many places. For example, the phrases "The outline of this paper is given as following..." "we starts", "we briafly explains", "we presents", "they satisfies", "entangled ones do not always violates"," both Cr(1) and Cr(2) does not..." are incorrect. Also, there are many typos in the manuscript like [[11]] in p.2, "bipartite" and "bi-partite" on p.2; caption to Fig. 2 (should be C_r^{(1)} in panel (c)) etc.

In short, I find that presented study has minor novelty as compared to Ref. [28] so that I cannot recommend its publication in Entropy.

Author Response

Dear referee

We do acknowledge the referee for his/her comments and the suggestions. We did modify our manuscript accordingly and the details are summarized below. We wish that all the issues are now settled down and our points become to be seen much clearer. 

Best, Corr. Author

Reply to Referee 2

Comment1: Although the problem (albeit it has been actually solved in Ref. [28], whereas the Bell-type correlations, though in the model without anisotropy, had been considered in Ref.[23], see also earlier work [12]) is interesting by itself, I consider the present manuscript to be of minor novelty. This is because all its components have been worked out earlier: the XXZ chain phase diagram in Ref. [28] of the manuscript, while quantum correlations (nonlocality) in the form of Bell inequalities in the above papers.”

Reply :

For the first, we do appreciate the comments by the second referee. In fact, it is correct that phase Diagrams of Spin-1 XXZ chain have been studied earlier by using the bosonization techniques Ref. [32] and numerical techniques (like DMRG method) Ref. [28]. However, here the novel feature of our work is to use the CGLMP correlation which has completely different meaning as a quantity that quantifies quantum phase in terms of nonlocalities.

In Ref. [12], the CHSH correlation is utilized in spin-1/2 XXZ model. On the contrary the CGLMP correlation is the generic version of the CHSH correlation for the arbitrary dimension of the local Hilbert space. Although the CGLMP correlation is firstly suggested in terms of the joint probability in Ref. [24], here we finally express it in terms of operator form, represents observable to be measured and apply it to the spin-1 chain. The spin models for S=1 have various phases in it and contains far more complex physical characteristics than those for S=1/2. We would like to stress that high dimensional Bell-type correlations are hardly applied to the many-body systems so far.

 Additionally, compared to the description of Bell correlation in terms of joint probabilities, the decomposition of our Bell operator Eq.(6) is advantageous in many regards. One can directly read off the structure of Eq.(6) and comprehend the Bell correlation as a linear combination of the first-order and second-order spin-spin correlations (see Eq.(9)). Remarkably, the role of the second-order correlation has never been addressed in the previous investigation of non-locality so far.

Through our numerical analysis, we stress the non-analyticity of the CGLMP correlation is found in the vicinity of the quantum phase transitions. Due to our description of the Bell operator Eq.(9), we can confirm that the critical behavior of the CGLMP correlation is attributed to the first-order and the second-order spin-spin correlations separately and .

References

[28] W. Chen et al., Phys. Rev. B 67 104401 (2003)

[23] A. L. Malvezzi et al., Phys. Rev. B 93 184428 (2016)

[12] L. Justino and T. R. Oliveira Phys. Rev. A 85 052128 (2012)

[24] D. Collins et al., Phys. Rev. Lett. 88 040404 (2002)

[32] H. J. Schulz, Phys. Rev. B 34 6372 (1986)

Comment2: “Minor remark is that the English is inadequate in many places.

Without the modification of our major claim, we revised our manuscript through putting emphasis on our main results while we correct the typos including the ones that is identified by the second referee. There is the list of corrections that had been attached below.

List of Corrections

  1. We remove the section ”MPS Representation and iDMRG method“ and Figure 1 “Graphical descriptions for the expectation value”, since it is too long.
    So, we add the brief explanation of the matrix product states at the beginning of section 3 “Quantum criticality through the correlation for the nonlocality tests”. (See line # ~ ) It is enough to understand our results.
  2. Since it is too long to explain the phases for the model Eq.(1), we reduced section 1 “The 1D spin-1 XXZ model with the on-site anisotropy”. (line #69~89)
  3. We revise some paragraphs from the beginning of section 3 to just before section 3.1 (line #121~138). After revision, we stress the novelty of our work in this part.
  4. Most of the 6th paragraph in section 0 “Introduction” has been deleted.
    It is because this paragraph is more suitable for section 4 “conclusion” and similar sentences already exists in the section “conclusion”. (line #69~79)
  5. In section 0 “Introduction”, the paragraph which contains the contents of recent papers on multipartite nonlocality in spin-1/2 chains has been deleted. The reason for it is that we have not dealt with multipartite Bell correlation in this work.
  6. We have inserted the sentence “The localizable entanglement, another measure of entanglement, is used to detect the phase transition in spin-1/2 XXZ model and to show the divergence of the entanglement length in valence-bond solid states”.
  7. “Especially, we analyze its Bell correlation in the spin-1 chain to identify its exotic quantum phase near the point of QPTs.”
    ⟹ “Then, we analyze its Bell correlation in the spin-1 chain to identify diverse quantum phase transitions. Moreover, we discuss whether violation of the Collins-Gisin-Linden-Massar-Popescu (CGLMP) inequality can be detected even in three-dimensional systems.
  8. “In sec.2, we present the Bell correlation in the CGLMP-Bell inequality from the generalized SLK-Mermin-Bell formalism described in an operator form of nonlocal order parameter.”
    ⟹ “In Sec.2, we show the Bell correlation in the CGLMP inequality from the generalized SLK-Mermin-Bell formalism described in an operator form.”
  9. “non-locality” ⟹ “nonlocality”
  10. “Sec. V” and “Sec. VI” ⟹ “Sec. 3” and “Sec.4”
  11. We have added the sentence “Our choice of model is made on the system with the versatile quantum phases occurred by the second-order magnetization term.” In section 1.
  12. After Eq.(5), we added the sentence “where the different choice of measurements on the second system, as shown in Eq.(4), results in the extra shift of the local phase.”
  13. “Thus, the operator can be simplified as ”

⟹ “Thus, together with the replacement, the Bell operator  can be rewritten in a simplified manner as”

  1. “This operator holds for the local basis of dimension d.”
    ⟹ “The operator can be applied to the system with the local basis dimension d.”
  2. “Such nonviolation of the CGLMP is consistent to the cases of CHSH in the earlier studies of 1D spin-1/2 XY chain and XXZ chain.”
    ⟹ “Such nonviolation of the CGLMP inequality is consistent to the cases of the CHSH inequality in the earlier studies of 1D spin-1/2 XY chain and XXZ chain.”

We corrected lots of sentences that are inappropriate for subject-verb agreement rules.

Reviewer 3 Report

In this submission the authors apply Bell-type correlation functions for analysis of quantum phase transitions in one-dimensional XXZ chain of S=1 spins. In my opinion the manuscript contains quite interesting technical results, that may represent certain interest. Nevertheless, I also think that this manuscript is not well written. It is too long, especially with consideration to the small amount of new results and but large number of quite secondary information. In addition, the manuscript is burdened a huge amount of abbreviations and with convoluted sentences. All this makes the reading quite difficult and boring. In my opinion the authors should make an effort to produce more concise version in the paper clearly stressing (along the text and not only in the conclusions) the crucial points and their relevance in the general context.
I would not recommend the present manuscript for publication but its amended version can be reconsidered.

Author Response

Dear referee

We do acknowledge the referee for his/her comments and the suggestions. We did modify our manuscript accordingly and the details are summarized below. We wish that all the issues are now settled down and our points become to be seen much clearer.

Best, Corr. Author

Reply to Referee 3

Comment1: “It is too long, especially with consideration to the small amount of new results and but large number of quite secondary information.”

Reply :

  • First of all, we acknowledge the comment by the referee. We would like to address that, following the suggestion given by the referee, we have removed large parts of our manuscript for the sake of brevity. They are summarized as following :
  1. We removed section “MPS representation and iDMRG method”. Instead, we have added the paragraph that briefly explains the matrix product states representation at the beginning of section “Quantum criticality through the correlation for the nonlocality tests”.
  2. In section 0 “Introduction”, the 5th paragraph which contains the contents of recent papers on multipartite nonlocality in spin-1/2 models has been deleted. The reason for it is that we have not dealt with multipartite Bell correlation in this work.
  3. Since being more suitable for section 4, the following sentences have been deleted in section 0 “Introduction”:
    “As a result, we found various QPTs that is accurately indicated by analyzing the derivatives of the CGLMP-SLK correlation : FM-to-XY, Haldane-to-AFM, Haldane-to-large-D, and XY1-to-XY2 quantum phase transitions. The first-order QPTs (FM-to-XY transitions) shows discontinuity of the CGLMP-SLK correlation. The Ising-type (Haldane-to-AFM and XY1-to-XY2) QPTs occur where the second derivative of the correlation vanishes. The Haldane-to-large-D transition, also referred to as the Gaussian phase transition, emerges when ∂⟨Br⟩/∂D = 0. The criticality of the correlation is caused by the fact that both the derivatives of the spin-spin correlations C(1) and C(2) are simultaneously zero in the criticality. Moreover, we discuss whether violation of the CGLMP-SLK inequality can be detected even in three-dimensional systems.”

The other parts that had been corrected were,

  1. In the section 1 “The 1D spin-1 XXZ model with on-site anisotropy” is redundant, we thought, that we give a concise account of the explanation on the quantum phases.
  2. Furthermore, in order to emphasize the key point in our work, we revise several paragraphs at the beginning of section 3. Therefore, we reduced secondary information in our literatures and put the emphasis on our main results more.

Comment2: “In addition, the manuscript is burdened a huge amount of abbreviations and with convoluted sentences.”

Reply :

  • We have reduced the number of abbreviations. We got rid of “AKLT”, “LMG”, “MPO”, and “DMRG”. Also, convoluted sentences were changed to simpler (see the list of corrections).

The complete list of corrections is given as following.

List of Corrections

  1. We remove the section ”MPS Representation and iDMRG method“ and Figure 1 “Graphical descriptions for the expectation value”, since it is too long.
    So, we add the brief explanation of the matrix product states at the beginning of section 3 “Quantum criticality through the correlation for the nonlocality tests”. (See line # ~ ) It is enough to understand our results.
  2. Since it is too long to explain the phases for the model Eq.(1), we reduced section 1 “The 1D spin-1 XXZ model with the on-site anisotropy”. (line #69~89)
  3. We revise some paragraphs from the beginning of section 3 to just before section 3.1 (line #121~138). After revision, we stress the novelty of our work in this part.
  4. Most of the 6th paragraph in section 0 “Introduction” has been deleted.
    It is because this paragraph is more suitable for section 4 “conclusion” and similar sentences already exists in the section “conclusion”. (line #69~79)
  5. In section 0 “Introduction”, the paragraph which contains the contents of recent papers on multipartite nonlocality in spin-1/2 chains has been deleted. The reason for it is that we have not dealt with multipartite Bell correlation in this work.

  6. We have inserted the sentence “The localizable entanglement, another measure of entanglement, is used to detect the phase transition in spin-1/2 XXZ model and to show the divergence of the entanglement length in valence-bond solid states”.
  7. “Especially, we analyze its Bell correlation in the spin-1 chain to identify its exotic quantum phase near the point of QPTs.”
    ⟹ “Then, we analyze its Bell correlation in the spin-1 chain to identify diverse quantum phase transitions. Moreover, we discuss whether violation of the Collins-Gisin-Linden-Massar-Popescu (CGLMP) inequality can be detected even in three-dimensional systems.
  8. “In sec.2, we present the Bell correlation in the CGLMP-Bell inequality from the generalized SLK-Mermin-Bell formalism described in an operator form of nonlocal order parameter.”
    ⟹ “In Sec.2, we show the Bell correlation in the CGLMP inequality from the generalized SLK-Mermin-Bell formalism described in an operator form.”
  9. “non-locality” ⟹ “nonlocality”
  10. “Sec. V” and “Sec. VI” ⟹ “Sec. 3” and “Sec.4”
  11. We have added the sentence “Our choice of model is made on the system with the versatile quantum phases occurred by the second-order magnetization term.” In section 1.
  12. After Eq.(5), we added the sentence “where the different choice of measurements on the second system, as shown in Eq.(4), results in the extra shift of the local phase.”
  13. “Thus, the operator can be simplified as ”

⟹ “Thus, together with the replacement, the Bell operator  can be rewritten in a simplified manner as”

  1. “This operator holds for the local basis of dimension d.”
    ⟹ “The operator can be applied to the system with the local basis dimension d.”
  2. “Such nonviolation of the CGLMP is consistent to the cases of CHSH in the earlier studies of 1D spin-1/2 XY chain and XXZ chain.”
    ⟹ “Such nonviolation of the CGLMP inequality is consistent to the cases of the CHSH inequality in the earlier studies of 1D spin-1/2 XY chain and XXZ chain.”
  3. We corrected lots of sentences that are inappropriate for subject-verb agreement rules.

Round 2

Reviewer 3 Report

The authors have satisfactorily answered my comments. Now I recommend the present manuscript for publication.